# Resection of Skin Cancer Resulting in Free Vascularized Tissue Reconstruction: Always a Therapeutic Failure?

**DOI:** 10.3390/cancers15092464

**Published:** 2023-04-25

**Authors:** Tina Rauchenwald, Angela Augustin, Theresa B. Steinbichler, Bernhard W. Zelger, Gerhard Pierer, Matthias Schmuth, Dolores Wolfram, Evi M. Morandi

**Affiliations:** 1Department of Plastic, Reconstructive and Aesthetic Surgery, Medical University of Innsbruck, 6020 Innsbruck, Tyrol, Austria; 2Department of Otorhinolaryngology, Head and Neck Surgery, Medical University of Innsbruck, 6020 Innsbruck, Tyrol, Austria; 3Private Praxis for Dermatopathology Innsbruck & Zams, 6020 Innsbruck, Tyrol, Austria; 4Department of Dermatology, Venerology and Allergology, Medical University of Innsbruck, 6020 Innsbruck, Tyrol, Austria

**Keywords:** skin cancer, perineural invasion, desmoplastic, dermatologic surgery, reconstructive surgery, free-flap reconstruction, basal cell carcinoma, cutaneous squamous cell carcinoma, melanoma

## Abstract

**Simple Summary:**

As the global burden of skin cancer rises, complex cases of locally advanced skin cancer are rising as well. Rare and aggressive dissemination mechanisms are often the cause for extensive skin cancer. Incohesive spreading patterns complicate correct diagnoses and may result in large tumors with multiple misguided treatment attempts. The aim of this study was to create awareness for cutaneous malignancies with rare, aggressive dissemination mechanisms to improve diagnostic and therapeutic strategies in such complex cases. We identified high-risk characteristics of locally advanced skin cancer warranting extended surgical treatment and a cohesive interdisciplinary approach.

**Abstract:**

The globally increasing incidence of cutaneous malignancies leads, in parallel, to increasing numbers of locally advanced skin cancer resulting in reconstructive surgery. Reasons for locally advanced skin cancer may be a patient’s neglect or aggressive tumor growth, such as desmoplastic growth or perineural invasion. This study investigates characteristics of cutaneous malignancies requiring microsurgical reconstruction with the aim of identifying possible pitfalls and improving diagnostic and therapeutic processes. A retrospective data analysis from 2015 to 2020 was conducted. Seventeen patients (*n* = 17) were included. The mean age at reconstructive surgery was 68.5 (±13) years. The majority of patients (14/17, 82%) presented with recurrent skin cancer. The most common histological entity was squamous cell carcinoma (10/17, 59%). All neoplasms showed at least one of the following histopathological characteristics: desmoplastic growth (12/17, 71%), perineural invasion (6/17, 35%), or tumor thickness of at least 6 mm (9/17, 53%). The mean number of surgical resections until cancer-free resection margins (R0) were achieved was 2.4 (±0.7). The local recurrence rate and the rate of distant metastases were 36%. Identified high-risk neoplastic characteristics, such as desmoplastic growth, perineural invasion, and a tumor depth of at least 6 mm, require a more extensive surgical treatment without concerns about defect size.

## 1. Introduction

Cutaneous malignancies are the most common malignancies in humans and are generally found in sun-exposed areas such as the face, neck, and scalp due to chronic UV exposure [1]. The majority of skin cancers are treated by simple surgical excision; however, the globally increasing incidence of cutaneous malignancies leads, in parallel, to higher numbers of locally advanced skin neoplasms requiring reconstructive surgery [2]. Locally advanced skin cancer may be caused by a patient’s neglect or, rarely, by highly invasive entities [3]. A patient’s neglect usually results in tumor growth over a lengthy period of time and may, in certain cases, be associated with abnormal illness behavior, depression, somatic anxiety disorder, or schizophrenia [4,5,6]. Wax et al. have previously categorized two groups of patients with extensive skin cancer requiring microsurgical, free vascularized tissue transfer [7,8]. While the first group consisted of patients with neglected skin cancer, the second group represented patients with multiple previous treatment attempts. An in-depth analysis of neoplastic entities was not performed in this study [7,8]. In patients presenting with multiple previous treatment attempts, rare and insidious mechanisms of dissemination, such as desmoplastic growth and perineural invasion, might be the cause for misguided diagnosis and, thus, treatment [9]. Correct diagnosis and treatment are often delayed due to the clinically concealed character of certain neoplastic entities and due to the difficulty of histopathological diagnosis [10]. Meanwhile, aggressive growth can result in the invasion of deeper structures such as cartilage or bone and, thus, require extensive surgery followed by microsurgical, free vascularized tissue reconstruction as the last option at the top of the reconstructive ladder [11,12]. As the global burden of locally advanced skin cancer is rapidly increasing, examples of treatment strategies in such complex cases are warranted.

This study investigates characteristics of cutaneous malignancies in the head and neck region that ultimately result in microsurgical, free vascularized tissue reconstruction. Moreover, we aim to create awareness for aggressive skin cancer dissemination mechanisms and to indicate possible pitfalls in patients’ treatment pathways by sharing our personal experience and approach in this work.

## 2. Materials and Methods

A retrospective data analysis of all patients who underwent microsurgical reconstruction with free vascularized tissue transfer after skin cancer resection in the head and neck region was conducted between 1 January 2015 and 31 December 2020 within our department. Data acquisition and management were performed according to the Standards of Good Scientific Practice and the Declaration of Helsinki in its present version (64th WMA General Assembly, Fortaleza, Brazil, 2013). The institutional ethics committee granted ethical approval (No. 1310/2021). Written and oral informed consent for data and image processing was obtained from all patients. The medical chart review included patient demographics, Fitzpatrick skin type, and history of previous skin cancer, of previous malignant disease, and of immunosuppression. Characteristics of neoplasms including localization, entity, histological details regarding desmoplastic growth, perineural invasion, tumor thickness, and depth of tissue infiltration (e.g., bone infiltration) were recorded. Oncological staging and preoperative imaging were documented. Surgical characteristics including the number of resections needed to achieve histopathologically confirmed cancer-free resection margins, the reconstructive technique, and postoperative complications were recorded. Resections until histopathologically confirmed cancer-free resection margins are achieved, followed by staged reconstruction, is pursued as the standard of care within our department in cases of locally advanced skin cancer. Postoperative complications were classified according to the Clavien–Dindo classification [13]. If one patient had more than one complication, the complication with the highest Clavien–Dindo grade was used for statistical analysis in the individual patient. Adjuvant therapy, recurrent disease, and follow-up were also documented.

### Statistical Analysis

Data were tabulated and stored in Microsoft Excel 16.16.21 (Microsoft, Redmond, DC, USA). Statistical analysis was performed using IBM SPSS Statistics 27 for Windows (IBM SPSS Software, Armond, NY, USA) and GraphPad Prism 8.4.3 (GraphPad Software LLC, La Jolla, CA, USA). Descriptive analysis data are presented as mean ± standard deviation (SD) unless stated otherwise.

## 3. Results

### 3.1. Patient Demographics

From 1 January 2015 to 31 December 2020, 17 patients (*n* = 17) fulfilled the inclusion criteria. All patients underwent microsurgical, free vascularized tissue reconstruction after a resection of locally advanced skin cancer in the head and neck region. All 17 patients were presented to an interdisciplinary tumor board and treatment was performed according to the tumor board’s decision. The mean age when reconstructive surgery occurred was 68.5 ± 13 years old. Eleven patients (11/17, 64.7%) were male and six patients (6/17, 35.3%) were female. The Fitzpatrick skin type was type 2 in all patients. The majority of patients (14/17, 82.4%) presented with a recurrent skin cancer and all of these patients (14/17, 82.4%) also had a history of prior skin malignancy at another localization. Three patients (3/17, 17.6%) had a history of previous malignant disease; one patient had carcinoma of the nasopharynx, another patient had non-Hodgkin’s lymphoma, and the third patient had lung cancer. Three other patients (3/17, 17.6%) received immunosuppressive medication; one patient for polycythemia vera and two patients after solid-organ transplantation.

### 3.2. Characteristics of Neoplasms

The most common localization was the scalp (6/17, 35.3%) and the most common histological entity was squamous cell carcinoma (SCC) (10/17, 58.8%). All neoplasms presented larger than 2 cm in diameter. Twelve neoplasms (12/17, 70.6%) were histopathologically diagnosed with desmoplastic growth and six neoplasms (6/17, 35.3%) with perineural invasion. Meanwhile, three neoplasms were histopathologically diagnosed with simultaneous desmoplastic growth and perineural invasion, and all neoplasms without these characteristics showed a tumor thickness of at least 6 mm (millimeters. The mean histological tumor thickness was 12.2 (±7.4) mm. Overall, clinical and histological bone infiltration was diagnosed in six cases (6/17, 35.3%), of which all but one showed desmoplastic growth. The clinical and histological characteristics are presented in Figure 1 and Table 1.

### 3.3. Oncological Staging and Imaging

Oncological staging was performed in 14 cases (14/17, 82.4%) during the current presentation and prior to microsurgical, free vascularized tissue reconstruction. Due to the low metastatic risk of basal cell carcinoma, staging was only performed in selected cases according to the most recent guidelines on basal cell carcinoma [14]; all cases without staging were cases diagnosed with basal cell carcinoma. Staging included radiological imaging, e.g., computed tomography (CT) (11/14, 78.6%), ultrasound of regional lymph nodes (11/14, 78.6%), positron emission tomography (PET) (2/14, 14.3%), and magnetic resonance imaging (MRI) (3/14, 21.4%). In more than half of the cases (8/14, 57.1%), staging imaging included ultrasound of regional lymph nodes alongside CT. During staging imaging, lymph node metastasis was suspected in five cases (5/17, 29.4%). Lymph node biopsy was performed in two of these cases and selective neck dissection was performed in the other three cases. Overall, lymph node biopsy was performed in four patients (4/17, 23.5%), of which two patients, as mentioned, were suspected of lymph node metastasis upon staging imaging and the other two patients each showed an enlarged lymph node intraoperatively, which was sent for histopathological diagnosis. In one patient, selective lymph node dissection was performed adjacent to a partial parotidectomy in a basal cell carcinoma of the auricular region. Overall, lymph node metastases were histologically confirmed in three patients (3/17, 17.6%), of which all showed previous suspicious imaging. One patient had a history of prior lymph node metastasis confirmed by histology. Distant metastasis of the lungs was suspected in two cases (2/17, 11.8%); one patient had scirrhous squamous cell carcinoma and one patient had adenocarcinoma.

Imaging of the head and neck by CT and/or magnetic resonance imaging (MRI) to investigate tumor depth and infiltration prior to surgical tumor resection was carried out in 14 patients (14/17, 82.4%).

### 3.4. Surgical Characteristics

Overall, the mean number of surgical resections prior to microsurgical, free vascularized tissue reconstruction was 2.2 (±0.8, range from 1 to 3). The mean number of days between the last surgical resection and staged reconstruction was 10 (±12.3) days. Immediate reconstruction was performed in seven cases (7/17, 41.2%) due to the extent of the defect created. Intraoperative frozen sections were analyzed in three of these cases. In four of these cases (4/17, 23.5%), the bone was so extensively infiltrated by the neoplasm that free resection margins could not be achieved despite bone resection. Due to the high morbidity due to resection and/or the anatomical impossibility of further resection, the interdisciplinary tumor board decided on adjuvant therapy in these cases. Two patients received adjuvant radiotherapy, one received adjuvant radiochemotherapy, and one received immunotherapy. The third patient, however, was diagnosed with lung cancer and was no longer eligible for surgical therapy escalation. In the remaining cases (13/17, 76.5%), the mean number of surgical resections until cancer-free resection margins (R0) were achieved was 2.4 (±0.7, range 1 to 3), which included the resection of infiltrated bone in three cases. The most frequently used free flap for reconstruction was the gracilis muscle flap combined with the split-thickness skin graft (8/17, 47.1%). The facial artery served as the recipient artery in 52.9% (9/17) of cases. One venous anastomosis was performed in all cases except one that required two venous anastomoses. Postoperative complications occurred in nine cases (9/17, 52.9%). All recorded complications were classified as Clavien–Dindo grade 3b. The most common complication was secondary hemorrhage at the recipient site (4/9, 44.4%). Total flap loss was observed in one patient (1/17, 5.9%). Surgical characteristics are presented in Table 2. One patient example is displayed in Figure 2.

### 3.5. Follow-Up

The mean follow-up time after reconstructive surgery was 25.3 (±20.9, range 0–63) months. Only one patient was lost to follow-up. Three patients died and four patients refused further treatment or follow-up. In nine cases, follow-up is still ongoing.

Overall, seven patients received adjuvant therapy (7/17, 41.2%) after microsurgical, free vascularized tissue transfer. Five patients underwent radiation therapy, one patient underwent radiochemotherapy, and one patient received immunotherapy with pembrolizumab.

Local recurrence occurred in 35.3% (6/17) of cases despite cancer-free resection margins (R0) being reached by histopathology. Local recurrence was present beneath the flap tissue; a tumor was found within the adjacent subcutaneous tissue in one case and skin lesions adjacent to the reconstructed tissue area were found in two cases. Progressive disease with the diagnosis of metastases was also observed in 35.3% of cases (6/17), of which two patients were diagnosed with novel distant metastases of the lung, two patients showed metastases after incomplete surgical resection (R1) (one patient showed an in-transit locoregional metastasis and the other patient showed bone metastasis), and two patients with known distant metastases of the lungs showed progression. Two of these patients were solid-organ transplant recipients under current immunosuppression. The mean time until the diagnosis of local recurrence was 7.7 (±13.9, range 0–36) months and the mean time until the diagnosis of metastases was 4.8 (±4.9, range 0–11) months.

Despite adjuvant therapy, local recurrence after adjuvant therapy occurred in 28.6% (2/7) of cases and progressive disease with metastases was observed in 57.1% (4/7) of cases. Of these six patients, one patient underwent surgical treatment again, two patients diagnosed with SCC and desmoplastic melanoma received immunotherapy with pembrolizumab, two patients diagnosed with BCC received immunotherapy with vismodegib, and one patient diagnosed with SCC after solid-organ transplantation received chemotherapy. Follow-up details are summarized in Table 3.

## 4. Discussion

General risk factors for cutaneous malignancies are well studied and documented in the literature. They include advanced age, Fitzpatrick skin type 1 and 2, and male sex [15,16,17,18]. The most important risk factor for the development of cutaneous neoplasms remains chronic UV damage [16,17,18]. While our patient group fits well into this scheme, specific risk factors for the development of locally advanced skin cancer resulting in microsurgical, free vascularized tissue reconstruction are still to be defined.

Immunosuppression is one risk factor known to be associated with more aggressive growth and higher rates of recurrence and metastasis [19,20]. Higher rates of cutaneous malignancies have been observed in solid-organ transplant recipients [21], in patients with human immunodeficiency virus (HIV) [22], and in hematopoietic stem cell transplant recipients [23]. While the risk of developing cutaneous neoplasms increases with the duration of immunosuppression [24,25], patients under immunosuppression generally also present with more invasive types of skin cancer [26,27]. For example, aggressive dissemination characteristics such as perineural invasion have been reported in up to 39% of solid-organ transplant recipients compared to a range from 2.5% to 14% of immunocompetent patients [19,28]. Furthermore, immunosuppression bares a higher risk of local recurrence and a higher rate of metastasis [29,30,31]. In certain cases, immunosuppression after organ transplantation has even been associated with skin cancer mortality that is up to nine times higher compared to that in the general population [32]. Interestingly, our patient group only included 17.6% of patients taking active immunosuppressive medication, suggesting further risk factors for aggressive cancer growth expanding beyond the clinically visible tumor appearance.

Strikingly, 82.4% of our patients presented with a locally recurrent lesion and had a history of prior skin malignancies. As locally recurrent skin cancers may already hint at aggressive neoplastic characteristics [33,34], subclinical infiltration should be considered before therapy, potentially yielding for extended resection margins. Recurrent neoplasms show a higher risk of further relapse and are also known to be associated with irregular dissemination mechanisms such as desmoplasia [35,36,37]. Desmoplasia, also often referred to as “scirrhous” or “sclerodermiform” growth, was first described in 1997 and is histopathologically characterized by “Indian filing”, a strand-like growth pattern of neoplastic cells [38]. In melanoma as well as non-melanoma skin cancer, desmoplastic growth is a known prognostic factor for local recurrence and for metastasis [39,40,41]. While desmoplasia has also been referred to as a risk factor for perineural invasion, a recent study argues that, in squamous cell carcinoma, perineural invasion is exclusively attributed to desmoplastic growth [42,43]. Perineural invasion is defined as neoplastic cells “in close proximity to the nerve” involving at least one third of the nerve’s circumference [44] and is diagnosed by immunohistochemistry [10].

In our study population, 88.2% were diagnosed with a cutaneous malignancy showing desmoplastic growth and/or perineural invasion surpassing the number of patients presenting with recurrent skin cancer already. The remaining patients without histological findings of desmoplastic growth or perineural invasion were diagnosed with cutaneous malignancies showing a histological tumor depth of at least 6 mm; all tumors included in this study had a diameter larger than 2 cm. A tumor depth of over 6 mm has been described as an independent prognostic factor for metastasis and recurrence [33]. At the same time, an increased depth and diameter may be associated with desmoplastic growth and, hence, perineural invasion, although the dependence between desmoplasia and perineural invasion has still not been fully explained [43].

Altogether, histological diagnosis referring to desmoplastic growth, perineural invasion, or a tumor depth of at least 6 mm should be alarming characteristics. Most of our patients necessitated at least two resections to achieve cancer-free resection margins and 35.3% developed local recurrence in only 7.7 (±13.9) months despite these cancer-free resection margins. Based on our findings, we suggest increased resection margins for cutaneous malignancies with these alarming characteristics that surpass the resection margins recommended by the guidelines. Interestingly, our study population included only one patient with desmoplastic melanoma, which might be attributed to the already larger resection margins applied in the therapy of melanoma [16,40]. Wider resection margins, complete histopathological margin analysis, and Moh’s surgery are recommended by the recent literature on complex skin cancer [45,46]. Nevertheless, Moh’s surgery and intraoperative frozen section control may be of limited value in lesions with desmoplastic growth or perineural invasion [47,48]. The incohesive spreading pattern of these aggressive dissemination mechanisms might not be displayed correctly on frozen sections and can, therefore, lead to a diagnostic discrepancy in a definitive histological diagnosis [49]. In these cases, immunohistochemical staining is the diagnostic tool of choice to detect desmoplastic growth and perineural invasion. Before performing microvascular, free vascularized tissue reconstruction, the histology ought to be definite, which demands staged reconstruction procedures with temporary defect coverage. Furthermore, in cases of suspected complex cutaneous malignancy, surgeons ought to prioritize cancer-free resection margins without concern for defect size. Additionally, imaging by MRI or CT early on should be considered to evaluate the infiltration of deeper structures prior to surgery.

Microvascular, free vascularized reconstruction in the head and neck represents a safe procedure and covers extended tissue defects. A good aesthetic and functional outcome is achieved provided that the defect analysis is carried out correctly and the adequate free flap is chosen for each individual application. The evaluation of complications in our patient collective showed only one total loss (5.9%) of a gracilis muscle flap, which was salvaged by secondary defect coverage using a deep inferior epigastric perforator (DIEP) flap. Increased morbidity in our patient collective originated from advanced age and the presence of underlying malignant diseases with a high prevalence of early local recurrence and distant metastases.

Regarding the local recurrence rate in this study, disease-free intervals before performing complex reconstructive surgery should be considered, and follow-up should be intensified and prolonged with a thorough evaluation of the surgical field in order to avoid the inconsiderate misinterpretation of recurrent lesions as a wound healing disorder. Additionally, adjuvant therapy with novel treatment options, e.g., biological agents, should be considered in these patients.

## 5. Conclusions

In conclusion, high-risk characteristics including desmoplastic growth, perineural invasion, or a tumor depth of at least 6 mm require a more aggressive surgical treatment. Resection margins should exceed those recommended for simple skin cancer excision, and resection ought to be carried out without concern for defect size. Although medical advancements offer a range of adjuvant therapy modalities, surgery still remains the best option for the curative treatment of primary skin cancer and microsurgical, free vascularized tissue reconstruction is a safe treatment option.

Thorough resection planning with the support of local imaging and an interdisciplinary approach with adequate follow-up regimens is necessary to achieve the best possible outcome for patients with locally advanced skin cancer and to diagnose aggressive dissemination mechanisms.

## Figures and Tables

**Figure 1 cancers-15-02464-f001:**
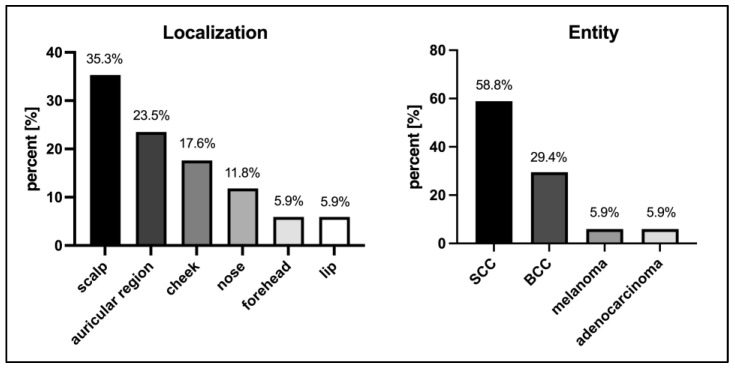
Characteristics of neoplasms. SCC—squamous cell carcinoma; BCC—basal cell carcinoma.

**Figure 2 cancers-15-02464-f002:**
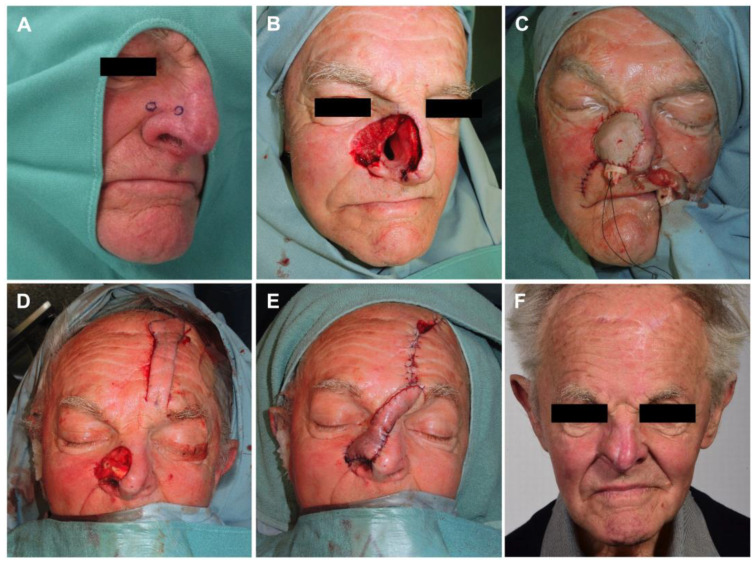
(**A**) Biopsies taken upon suspicious skin changes; (**B**) resection defect after tumor-free resection margins reached; (**C**) first-step reconstruction of nasal inner lining with free radial forearm flap; (**D**) second-step reconstruction with nasal alar reconstruction by autologous cartilage graft and (**E**) pedicled paramedian forehead flap; (**F**) follow-up result 18 months postoperatively.

**Table 1 cancers-15-02464-t001:** Histological characteristics of neoplasms.

Histological Characteristic	Entity	Number of Patients	Percentage (%)
Desmoplastic growth	Overall (*n* = 17)	12/17	70.6%
	SCC ^1^ (*n* = 10)	6/10	60.0%
	BCC ^2^ (*n* = 5)	5/5	100%
	Melanoma (*n* = 1)	1/1	100%
	Adenocarcinoma (*n* = 1)	0/1	0%
Perineural invasion	Overall (*n* = 17)	6/17	35.3%
	SCC ^1^ (*n* = 10)	5/10	50.0%
	BCC ^2^ (*n* = 5)	1/5	20.0%
	Melanoma (*n* = 1)	0/1	0%
	Adenocarcinoma (*n* = 1)	0/1	0%
Infiltration of bone	Overall (*n* = 17)	7/17	41.2%
	SCC ^1^ (*n* = 10)	3/10	30.0%
	BCC ^2^ (*n* = 5)	3/5	60.0%
	Melanoma (*n* = 1)	1/1	100%
	Adenocarcinoma (*n* = 1)	0/1	0%

^1^ SCC—squamous cell carcinoma; ^2^ BCC—basal cell carcinoma.

**Table 2 cancers-15-02464-t002:** Surgical characteristics.

Surgical Characteristic		Number of Patients	Percentage (%)
Free flap	Gracilis muscle flap	8/17	47.1%
	Radial forearm flap	4/17	23.5%
	Latissimus dorsi muscle flap	3/17	17.6%
	Anterolateral thigh flap	2/17	11.8%
Recipient artery	Facial artery	9/17	52.9%
	Superficial temporal artery	4/17	23.5%
	Superior thyroid artery	3/17	17.6%
	Ascending pharyngeal artery	1/17	5.9%
Complications	Hematoma (recipient site)	4/17	23.5%
	Hematoma (donor site)	1/17	5.9%
	Partial flap necrosis	1/17	5.9%
	Total flap necrosis	1/17	5.9%
	Wound infection (donor site)	1/17	5.9%
	Microvascular (arterial thrombus)	1/17	5.9%

**Table 3 cancers-15-02464-t003:** Follow-up.

	Number of Patients	Percentage (%)	Months
Local recurrence after reconstructive surgery	6/17	35.3%	-
Progressive disease with metastases after reconstructive surgery	6/17	35.3%	-
Local recurrence after adjuvant therapy	2/7	28.6%	-
Progressive disease with metastases after adjuvant therapy	4/7	57.1%	-
Time until local recurrence	-	-	7.7 (±13.9)
Time until progressive disease	-	-	4.8 (±4.9)

## Data Availability

Data supporting the findings of this study are available from the authors upon request.

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
