# Peer review of "Resection of Skin Cancer Resulting in Free Vascularized Tissue Reconstruction: Always a Therapeutic Failure?"

_cancers, 2023, doi:10.3390/cancers15092464_

Round 1
Reviewer 1 Report
I would like to congratulate the authors to their results and article. There are a few question that I would like to ask.
First, the authors recommend to increase the surgical margin in these aggressive, often recurrent skin lesions. Do they have any scientific data to support this statement? Were all the recurrencies in the scar tissue around the flap? Is field cancerization an issue, especially in the scalp? Since you performed free flaps in all 17 cases, the defect size matters probably less than if you plan to reconstruct with local flaps. Did you see a difference in tumor free margins in pathologic work-ups, i.e. wasn’t there already an increase in safety margin, also since you used more than 1 resection (2.24 in average)?
6 of 17 patients had suspected or actual metastases. Did that influence your planning? Was there an interdisciplinary tumor board involved in all these cases?
Last but not least, the percentages indicated (e.g. 2.24 resections in average) suggest a mathematical precision beyond the 17 patients involved.
Reviewer 2 Report
please elaborate on the low number of patients that got to free flap among all of your patients, please elaborate on the low percentage of adjuvant therapy , which is standard, and discuss the use of biological agents.
